# SSCNN-S: A Spectral-Spatial Convolution Neural Network with Siamese Architecture for Change Detection

**Tianming Zhan [1,2], Bo Song [2], Yang Xu [3], Minghua Wan [1,2], Xin Wang [2], Guowei Yang [2,4] and Zebin Wu [3,*]**

1   Jiangsu Key Construction Laboratory of Audit Information Engineering, Nanjing Audit University,
    Nanjing 211815, China; ztm@nau.edu.cn (T.Z.); 270223@nau.edu.cn (M.W.)
2   School of Information Engineering, Nanjing Audit University, Nanjing 211815, China;
    mg1909003@stu.nau.edu.cn (B.S.); 209204@nau.edu.cn (X.W.); 270178@nau.edu.cn (G.Y.)
3   School of Computer Science and Engineering, Nanjing University of Science and Technology,
    Nanjing 210094, China; xuyangth90@njust.edu.cn
4   School of Electronic Information, Qingdao University, Qingdao 266071, China
*   Correspondence: wuzb@njust.edu.cn

**Abstract:** In this paper, a spectral-spatial convolution neural network with Siamese architecture (SSCNN-S) for hyperspectral image (HSI) change detection (CD) is proposed. First, tensors are extracted in two HSIs recorded at different time points separately and tensor pairs are constructed. The tensor pairs are then incorporated into the spectral-spatial network to obtain two spectral-spatial vectors. Thereafter, the Euclidean distances of the two spectral-spatial vectors are calculated to represent the similarity of the tensor pairs. We use a Siamese network based on contrastive loss to train and optimize the network so that the Euclidean distance output by the network describes the similarity of tensor pairs as accurately as possible. Finally, the values obtained by inputting all tensor pairs into the trained model are used to judge whether a pixel belongs to the change area. SSCNN-S aims to transform the problem of HSI CD into a problem of similarity measurement for tensor pairs by introducing the Siamese network. The network used to extract tensor features in SSCNN-S combines spectral and spatial information to reduce the impact of noise on CD. Additionally, a useful four-test scoring method is proposed to improve the experimental efficiency instead of taking the mean value from multiple measurements. Experiments on real data sets have demonstrated the validity of the SSCNN-S method.

**Keywords:** spectral-spatial combination; hyperspectral image (HSI); change detection (CD); Siamese network

## 1. Introduction

Due to the development of remote sensing technology it is possible to obtain hyperspectral images (HSIs) of the same area at different time points. Change detection (CD) using multitemporal remote sensing data has an important application value in disaster assessment [1], terrain change analysis [2], urban change analysis [3] and resource auditing. The rich spectral and spatial information of HSIs, which contain hundreds of bands, provides a more powerful data source for object observation. In [4], the author divides CD into the following categories: anomaly detection [5–7], binary and multiclass CD [8–11] and CD based on time series data [12,13].

Many researchers have studied the multispectral CD task with a low number of bands and proposed a few CD algorithms. Change vector analysis (CVA) [14] is often combined with other methods. By calculating the spectral change vector corresponding to a pixel, the magnitude and angle of the spectral change of the pixel are analyzed. Multivariate alteration detection (MAD) [15] and iteratively reweighted multivariate alteration detection (IR-MAD) [16] are based on canonical correlation analysis (CCA) [17]. The change area is determined by calculating the values and their weights of MAD variables. In addition, it is

also a feasible strategy to classify HSIs at different times and to compare the classification results to determine the change area. This strategy can introduce excellent algorithms in the field of HSI classification into CD [18,19]. The aforementioned methods use algebraic and statistical theories to extract the features of the spectral vector or spectral change vector and have achieved good results with low-dimensional space.

However, the CD algorithm—which is suitable for low-dimensional space—does not work well in the high-dimensional space of HSIs [20,21]. One important reason for this is that the limited calculation accuracy of computers will cause certain calculation errors. While these calculation errors have a limited impact on the final result in low-dimensional space, a few vector and matrix operations performed in high-dimensional space (e.g., solving the inverse matrix and eigenvectors of high-dimensional matrices) may be greatly affected by calculation errors.

Additionally, due to the strong correlation between adjacent bands in HSIs, a large amount of redundant information also increases the difficulty of feature extraction [22–24]. Therefore, reducing the dimension of the HSI is an important topic. The aforementioned CCA algorithm achieves the goal of dimension reduction by finding two typical variables of a lower dimension to represent the original vector (of a higher dimension). Principal component analysis (PCA) [25] solves the eigenvalues and eigenvectors of the covariance matrix and selects the eigenvectors corresponding to the largest $k$ eigenvalues to form a linear transformation matrix to map the original data to the specified dimension $k$. Considering the high correlation between adjacent hyperspectral bands, it is also prudent to select a certain number of bands from the original hyperspectral image for feature extraction. Ma et al. [26] improved the effect of CD by selecting bands with more change information and processing them to suppress noise. In addition to linear dimension reduction, manifold learning as a non-linear dimension reduction method is also applied in hyperspectral image processing. Yu et al. [27] improved the neighborhood rough set, proposed the local neighborhood selection method combined with local linear embedding (INRSLLE) and effectively reduced the dimension by using local manifold learning. With the development of deep learning [28–30], it has been applied to reduce the dimension and extract the features [31–33]. Chen et al. [34] constructed a 1D convolution network to extract the features of the spectral vectors corresponding to a single pixel and achieved the effect of hyperspectral dimension reduction. Lv et al. [35] suppressed noise in synthetic aperture radar (SAR) images by stacking contractive autoencoder (sCAE) and extracting features to improve the accuracy of CD.

Noise is unavoidable in hyperspectral images and originates from the internal noise of the hyperspectral imager itself and external factors such as atmospheric scattering. Li et al. [36] mentioned that CD algorithms that only extract the spectral information of pixels will be affected by noise, thereby resulting in poor CD results. Therefore, it is important to extract HSI features other than spectral features. Wang et al. [37] introduced the endmember abundance information obtained by unmixing into an affinity matrix and used a convolution neural network (CNN) for CD to achieve good results. Spatial information is an important source of information for HSIs. Notably, a few advanced algorithms have introduced the extraction of spatial features. Wang et al. [38] used 1D and 2D convolution networks to extract the spectral and spatial features of HSIs, respectively. Huang et al. [39] proposed the tensor-based hyperspectral remote sensing images underlying features change information model (TFS-Cube) for feature extraction, which also included spatial information. Furthermore, Ran et al. [40] extracted neighborhood spatial information using three different combinations of filters. Roy et al. [41] proposed a hybrid spectral CNN (HybridSN) to blend 2D convolution networks with 3D convolution networks for more abstract spatial features.

Tensor-based methods are widely used in HSI processing to extract spectral and spatial information simultaneously. Similar to CVA, we can obtain the change tensor by subtracting two tensors. However, this results in losing the original spatial information of the HSI. While we can also classify the two tensors separately and determine whether the pixel belongs to the change area based on the classification results, this requires prior knowledge of which categories the tensor can be classified into.

Based on the face and action recognition task [42–46], we can segment the change area using a Siamese network. A Siamese network measures the similarity of tensors and can accept two inputs. The more similar the two inputs, the smaller the corresponding output value and vice versa. A Siamese network can solve the aforementioned problems: it can not only retain complete spatial information but also does not require any prior information about categories.

Therefore, a spectral-spatial network with Siamese architecture (SSCNN-S) is proposed to solve the problem of hyperspectral binary CD. First, for each pixel to be detected, the spectral information of the pixel and its neighborhoods are extracted from the two HSIs recorded at different time points to form a tensor pair as the input of the network. The vectors in the tensor pair are then incorporated into a Siamese network composed of the spectral module and the spatial module to obtain the corresponding spectral-spatial vectors. The Euclidean distance of the two spectral-spatial vectors is used as the similarity of the two tensors in the tensor pair. After obtaining the similarity for each pixel, the similarity is binarized by the threshold method to generate the final CD result.

The method proposed in this paper uses the theory of deep learning to measure the similarity of high-dimensional tensors and detect changes, which has the following main advantages. Using the Siamese network, the problem of CD is transformed into a problem of measuring the similarity of two tensors, which retains the complete spectral and spatial information of HSIs compared with the differential method. Extracting the spectral characteristics of the tensor through a 1D convolution network and reducing the dimensions in the Siamese network can effectively reduce the parameters of the network and increase the CD speed. Moreover, a 2D convolution network extracts spatial characteristics from the reduced dimension tensor, which can reduce the impact of noise and improve the CD accuracy. Experiments using three real data sets show that the SSCNN-S method proposed in this paper shows good performance in solving the problem of CD in HSIs. The main contributions of this work can be summarized as follows:

(1) 1D and 2D convolutional neural networks are used to extract spectral features and spatial features while local tensors are converted into spectral-spatial vectors. In this manner, the spectral and the spatial features are combined to increase detection speed.
(2) The introduction of the Siamese network helps to retain the original spatial features and can introduce advanced hyperspectral classification methods [47] into CD without the prior information of the number of categories and other processing methods [48].
(3) The four-test scoring method is proposed. This method is mainly used in parameter selection experiments with uncertain results. For each set of parameters, the method can give the final results based on the results of two to four independent experiments as the basis for parameter selection.

The other parts of this paper are arranged as follows. The second part details the proposed hyperspectral CD method (SSCNN-S) based on the Siamese network and the spectral-spatial combination method. The third part describes the experiments performed in this study by outlining the utilized data set and evaluation index as well as comparing and analyzing existing algorithms. The final part summarizes this paper.

## 2. Materials and Methods

### 2.1. Establishing the Sample Set

Let $X^{(t)} \in \mathbb{R}^{h \times w \times c}$ represent the HSI at the $t^{th}$ time where $h$ and $w$ correspond to the height and width of the hyperspectral spatial dimension, respectively, and $c$ represents the number of bands in the HSI $t \in \{1, 2\}$. Considering the $i^{th}$ pixel in $X^{(1)}$, to extract both the spectral and spatial features of this pixel, we extract the spectral information of this pixel and its neighborhood to form a hyperspectral tensor $X_i^{(1)} \in \mathbb{R}^{b \times b \times c}$ where $b$ represents the spatial dimension of the hyperspectral tensor. The corresponding tensors from each pixel in $X^{(1)}$ and $X^{(2)}$ are extracted to form a tensor set $\mathcal{X}^{(1)} = \left\{ X_1^{(1)}, X_2^{(1)}, \ldots, X_{hw}^{(1)} \right\}$ and $\mathcal{X}^{(2)} = \left\{ X_1^{(2)}, X_2^{(2)}, \ldots, X_{hw}^{(2)} \right\}$. To easily use the Siamese network to measure the similarity between tensors, we need to pair the tensors corresponding to the same pixels in $\mathcal{X}^{(1)}$ and $\mathcal{X}^{(2)}$ to form a tensor pair set $\mathcal{X} = \left\{ \left( X_1^{(1)}, X_1^{(2)} \right), \left( X_2^{(1)}, X_2^{(2)} \right), \ldots, \left( X_{hw}^{(1)}, X_{hw}^{(2)} \right) \right\}$. Label $\mathcal{Y} = \{ Y_1, Y_2, \ldots, Y_{hw} \}$ corresponds to $\mathcal{X}$ where:

$$Y_i = \begin{cases} 0 & \text{the } i^{th} \text{ pixel is unchanged} \\ 1 & \text{the } i^{th} \text{ pixel is changed} \end{cases} \tag{1}$$

Thus, $(\mathcal{X}, \mathcal{Y})$ can be considered as the sample set. The sample set is randomly divided into the training set $(\mathcal{X}_{train}, \mathcal{Y}_{train})$, validation set $(\mathcal{X}_{val}, \mathcal{Y}_{val})$ and test set $(\mathcal{X}_{test}, \mathcal{Y}_{test})$.

### 2.2. Extract Spectral-Spatial Features of the Hyperspectral Tensor

Similar to a previous work [38], we extracted the spectral and spatial features of the hyperspectral tensor $X_i^{(t)}$ using a 1D and 2D convolution neural network (Figure 1). Unlike the classification task, the network does not output a specific category but a spectral-spatial vector that combines spectral and spatial features. The network is divided into a spectral module and a spatial module where the spectral module is used to extract the spectral features of hyperspectral tensors while reducing the dimension of the tensors and the spatial module is used to extract the spatial features of the reduced tensors.

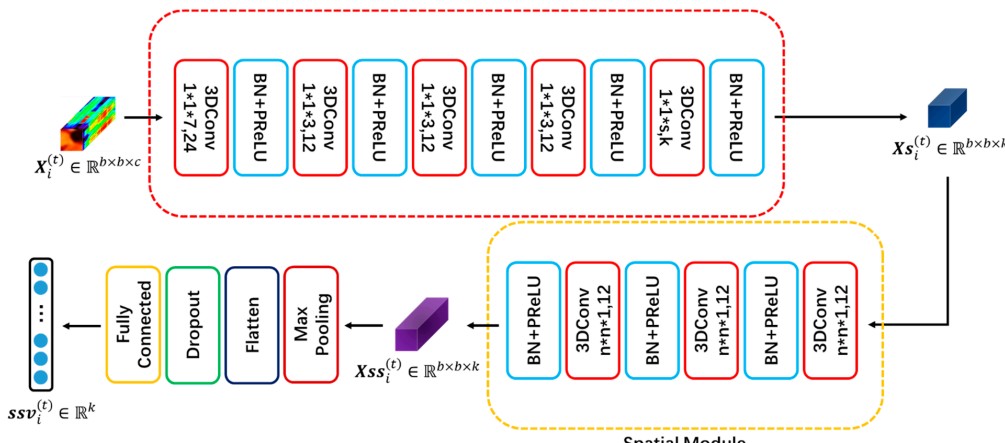

**Figure 1.** Network for extracting spectral-spatial features of hyperspectral tensors.

### 2.2.1. Spectral Module

HSIs have abundant spectral information. Most methods consider the extraction of spectral features as an important research topic. A hyperspectral tensor $X_i^{(t)} \in \mathbb{R}^{b \times b \times c}$ contains $b \times b$ spectral vectors and each has a dimension of $c$. Therefore, a 1D convolutional network can be used to extract the features of each spectral vector [34] and combine them to obtain the spectral features of the tensor. Another important role of 1D convolution is to reduce the spectral dimension of the tensor to $k$. In each convolution layer, different convolution kernels correspond to different methods of dimension reduction and these dimension reduction parameters can be automatically learned during network training. This can reduce the number of model parameters, which save computing resources and storage space while effectively improving the model training speed.

In network training, the overall distribution of the activation function value of the hidden layer shifts greatly, which causes the gradient to disappear and the training speed to decrease. To avoid this problem, we used batch normalization (BN) [49] in the spectral module to force the overall distribution back to the standard normal distribution.

The activation function used in the spectral module is the parametric rectified linear unit (PReLU) [50] whose function expression is as follows:

$$\text{PReLU}(x_p) = \max\{0, x_p\} + a_p \min\{0, x_p\} \tag{2}$$

where $x_p$ represents the input of the activation function for the $p$th channel and $a_p$ is a very small positive number. While $a_p$ is not set artificially, it is updated during model training as follows:

$$\Delta a_p \leftarrow m \Delta a_p + lr \frac{\partial loss}{\partial a_p} \tag{3}$$

where $m$ represents momentum and $lr$ represents the learning rate. HSIs contain values less than 0. Compared with the rectified linear unit (ReLU) function [51], the PReLU function does not set the activation values less than 0 directly to 0. Instead, it compresses them to a close negative value of 0. Although a few parameters that require training are added, faster model training is more advantageous.

### 2.2.2. Spatial Module

Although the spatial resolution of the HSI is not high, extracting the spatial features of the HSI as an adjunct basis for CD can help reduce the interference of noise in the spectral information. After dimension reduction, the hyperspectral tensor $Xs_i^{(t)} \in \mathbb{R}^{b \times b \times k}$ contains $k$ tensors of size $b \times b$, each with a large amount of spatial information [52]. We extract spatial information [34] using a 2D convolution network with a kernel size of $n \times n$. In the spatial module, we still use PReLU and BN after each convolution layer.

### 2.2.3. Achieve Spectral-Spatial Vector

After the spatial module, a hyperspectral tensor $Xss_i^{(t)} \in \mathbb{R}^{b \times b \times k}$ is obtained that combines the spectral and spatial features. $Xss_i^{(t)}$ is then transformed into a vector input into the fully connected layer through pooling, flattening and dropout. The output vector $ssv_i^{(t)} \in \mathbb{R}^k$ is used as the spectral-spatial vector of the input tensor.

### 2.3. Contrastive Loss in the Siamese Network

Consider the $i^{th}$ tensor group in tensor set $\left( X_i^{(1)}, X_i^{(2)} \right)$ whose corresponding spectral-spatial vectors are $ssv_i^{(1)}$ and $ssv_i^{(2)}$ (labeled $Y_i$). To describe the similarity between $ssv_i^{(1)}$ and $ssv_i^{(2)}$, we calculate the Euclidean distance between them:

$$dist\left( ssv_i^{(1)}, ssv_i^{(2)} \right) = \| ssv_i^{(1)} - ssv_i^{(2)} \|_2 \tag{4}$$

We want the $dist\left(ssv_i^{(1)}, ssv_i^{(2)}\right)$ to be as large as possible when $Y_i = 1$ and the $dist\left(ssv_i^{(1)}, ssv_i^{(2)}\right)$ to be as small as possible when $Y_i = 0$. Therefore, we use contrastive loss [51] as the loss function for model optimization:

$$Loss = \frac{1}{N}\sum_{i=1}^{N}\left[(1-Y_i)dist\left(ssv_i^{(1)}, ssv_i^{(2)}\right) + Y_i max\left\{m - dist\left(ssv_i^{(1)}, ssv_i^{(2)}\right), 0\right\}\right] \quad (5)$$

where $N$ is the number of training samples and $m$ is a boundary value that is set artificially. When $Y_i = 0$, the loss function becomes $dist\left(ssv_i^{(1)}, ssv_i^{(2)}\right)$. Only $ssv_i^{(1)}$ and $ssv_i^{(2)}$ are close enough to decrease the loss. When $Y_i = 1$, the loss function becomes $max\left\{m - dist\left(ssv_i^{(1)}, ssv_i^{(2)}\right), 0\right\}$. Only $ssv_i^{(1)}$ and $ssv_i^{(2)}$ are far enough to decrease the loss. When the $dist\left(ssv_i^{(1)}, ssv_i^{(2)}\right) > m$, loss is regarded as 0, which limits the influence of the tensor pair with too large a distance on overall loss.

### 2.4. Proposed SSCNN-S Method

The CD method of SSCNN-S is presented in Figure 2. SSCNN-S trains a Siamese network using training sets $(\mathcal{X}_{train}, \mathcal{Y}_{train})$. During training, the features of each tensor pair in $\mathcal{X}_{train}$ are extracted through the feature extraction network to obtain the corresponding spectral-spatial vector and the Euclidean distances of the two spectral-spatial vectors are calculated as the output of the network.

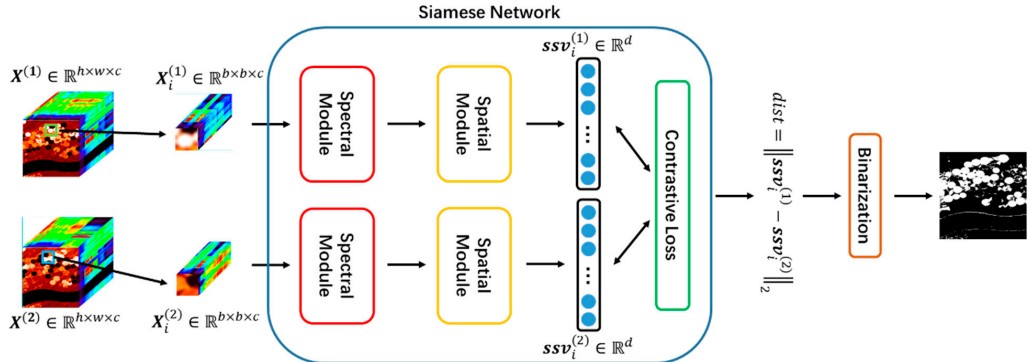

**Figure 2.** Overview of the spectral-spatial convolution neural network with Siamese architecture (SSCNN-S).

To segment the change region, we also need to find a binarization method that converts the output distance to 0 or 1. In SSCNN-S, we use a threshold-based binarization method; that is, to find a threshold $\theta$ for each $dist$ we use the binarization function:

$$Bin(dist, \theta) = \begin{cases} 0 & dist \le \theta \\ 1 & dist > \theta \end{cases} \quad (6)$$

Notably, the threshold $\theta$ can be determined in several ways. In SSCNN-S, we select the best threshold by traversing all possible thresholds in the validation set. This method is chosen because neither the validation set nor the test set participate in model training while the randomness of the validation set and test set generation allows them to be regarded as having approximately the same distribution. After binarizing the distances for each pixel, a CD result map can be obtained.

The detailed algorithm of SSCNN-S is shown in Algorithm 1.

---

**Algorithm 1:** Algorithm of SSCNN-S for hyperspectral image (HSI) change detection (CD).

**Input:**

1. Two HSIs of the same region at different times with ground truthing.

2. The number of training pairs $N_t$ and the number of validation pairs $N_v$.

**Step 1:** Construct the corresponding tensor sets $\mathcal{X}^{(1)}$ and $\mathcal{X}^{(2)}$ for two HSIs and pair them to form a tensor pair $\mathcal{X}$ and generate the sample set $(\mathcal{X}, \mathcal{Y})$ according to the change situation reflected by the ground truthing.

**Step 2:** Randomly select $N_t$ pairs in $\mathcal{X}$ as the training set $G_{train}$ and randomly select $N_v$ pairs in $\mathcal{X}$ as the validation set $G_{validation}$.

**Step 3:** Input $G_{train}$ and $G_{validation}$ to the network.

**Step 4:** Train the model and obtain the optimal parameters.

**Step 5:** Traverse all possible thresholds in the validation set to select the optimal threshold $\theta$.

**Step 6:** Calculate the distance for each pixel. If the distance is greater than $\theta$, it is considered as a changed pixel; otherwise, it is considered an unchanged pixel.

**Output:**

1. Change map.

---

## 3. Results

To verify the effect of SSCNN-S on CD, we first introduce three real hyperspectral data sets used for experiments and then provide indexes for evaluating the effects of different algorithms. Finally, we provide the experimental results and corresponding analyses on each data set.

### 3.1. Data Sets

We selected three real hyperspectral data sets (Figure 3): Farmland [53], River [37] and USA [54]. All three data sets were collected using Earth Observing-1 (EO-1) Hyperion data. The EO-1 hyperspectral imager covers electromagnetic waves with wavelengths ranging from 0.4 to 2.5 micrometers. EO-1 has a spectral resolution of 10 nm and a spatial resolution of about 30 m, with a total of 242 different bands.

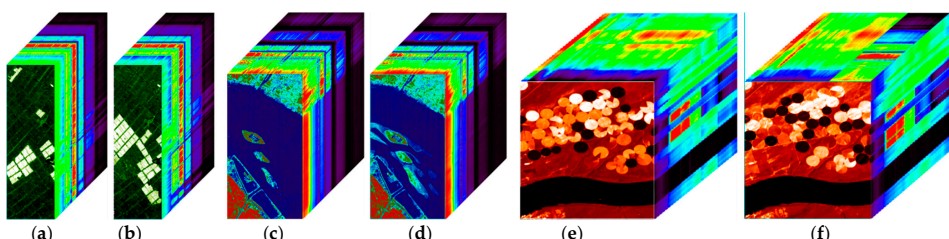

|(a)|(b)|(c)|(d)|(e)|(f)|

**Figure 3.** Experimental data sets: (**a**) Farmland data set on 3 May 2006. (**b**) Farmland data set on 21 April 2006. (**c**) River data set on 3 May 2013. (**d**) River data set on 31 December 2013. (**e**) USA data set on 1 May 2004. (**f**) USA data set on 8 May 2007.

The first data set, Farmland, was selected from a farmland area in Yancheng, Jiangsu Province, China. The data set primarily depicted changes in cultivated land. The time points of the two hyperspectral images were 3 May 2006 and 23 April 2007. Their spatial size was $450 \times 140$ pixels with 155 bands after removing low signal-noise-ratio (SNR) bands. Referring to the number of samples set in [37], we randomly selected 13,200 pixels as the training set of which 4400 were changed pixels and 8800 were unchanged pixels. Additionally, 6600 pixels were randomly selected as the verification set.

The second data set, River, was selected from a river region in Jiangsu Province, China. This data set mainly reflected material changes in the river. The selected time points were 3 May 2013 and 31 December 2013. The spatial size was $463 \times 241$ pixels with 198 bands. Referring also to the settings in [37], 3750 pixels were randomly selected as the training set of which 1250 were changed pixels and 2500 were unchanged pixels. Additionally, 1875 pixels were randomly selected as the verification set.

The third data set, USA, was from irrigated farmland in Hermiston, Umatilla County, Oregon, USA. It covered soil, irrigation areas, rivers and other terrain. The spatial size of the images was 307 × 241 pixels with 154 bands. Referring to the settings in [54], we randomly selected 7232 pixels as the training set of which 3313 were changed pixels and 3919 were unchanged pixels. A total of 3616 pixels were used for the validation sets.

### 3.2. Evaluation Index

We evaluated the difference between the CD result of the algorithm and the ground truth value to evaluate the results. For a binary CD problem, we supposed the total number of pixels to be tested was $T$. There were only one of four possible scenarios for each pixel: correctly classifying the changing pixels, whose number was denoted as $TP$; incorrectly classifying the changing pixels, whose number was denoted as $FP$; correctly classifying the changing pixels, whose number was denoted as $TN$; incorrectly classifying the changing pixels, whose number was denoted as $FN$. Then we obtained:

$$T = TP + FP + TN + FN \tag{7}$$

The first index introduced was overall accuracy (OA), which was calculated by:

$$OA = \frac{TP + TN}{T} \tag{8}$$

The $OA$ was used to measure the proportion of pixels correctly classified by the algorithm. In this index, $TP$ and $TN$ had the same impact on the $OA$.

However, in the case of extremely unbalanced data sets, only using the $OA$ as an evaluation index was problematic. Taking the River data set as an example, it contained 111,583 pixels. According to the ground truth, the actual percentage of changed pixels was not greater than 10%. This implied that if a model classified all of the pixels as unchanged, then the $OA$ would exceed 90%; however, this was not the model we required. Therefore, we introduced a second evaluation index, a Kappa coefficient, which was calculated as follows:

$$Kappa = \frac{OA - p_e}{1 - p_e} \tag{9}$$

where $p_e$ could be calculated by:

$$p_e = \frac{(TP + FP)(TP + FN)}{T^2} + \frac{(TN + FP)(TN + FN)}{T^2} \tag{10}$$

Effectively introducing the Kappa coefficient solved the problem of the consistency of model predictions. The relatively small number of changed pixels in the three data sets implied that the Kappa coefficient imposed greater penalties on the FP.

### 3.3. Experimental Results

All of the experiences in this paper were carried out on a personal computer that was equipped with an Intel Core i7-9700K Central Processing Unit (CPU) and an independent Graphics Processing Unit (GPU) of NVIDIA GeForce RTX 2080. We set the batch size to 128. We chose the Adam optimizer and its parameters were the default parameters. The boundary value $m$ in contrastive loss was set to 1.5. Due to the randomness of SSCNN-S, we ran it four times on each data set to calculate the mean and standard deviation of each index as the final result. To analyze the effect of the spatial module in CD, the method of extracting spectral features $Xs_i^{(t)}$ obtained after dimension reduction for CD was also included in the experiment, which was the spectral convolution neural network with Siamese architecture (SCNN-S).

To evaluate the effectiveness of SSCNN-S, we selected several existing algorithms for comparison: CVA [14], PCACVA [25], a support vector machine (SVM) [19,55], patch-based CNN (PBCNN) [56], Hybrid Spectral CNN (HybridSN) [41] and GETNET [37], which used

different remote sensing image CD strategies and techniques. Figures 4–6 intuitively show the segmentation results of the different algorithms on the three sets of data sets. Table 1 quantitatively compares the effects of various algorithms using evaluation indexes. In order to evaluate the time complexity of each algorithm, in Table 2, we provide the running time of different algorithms on the three data sets for reference.

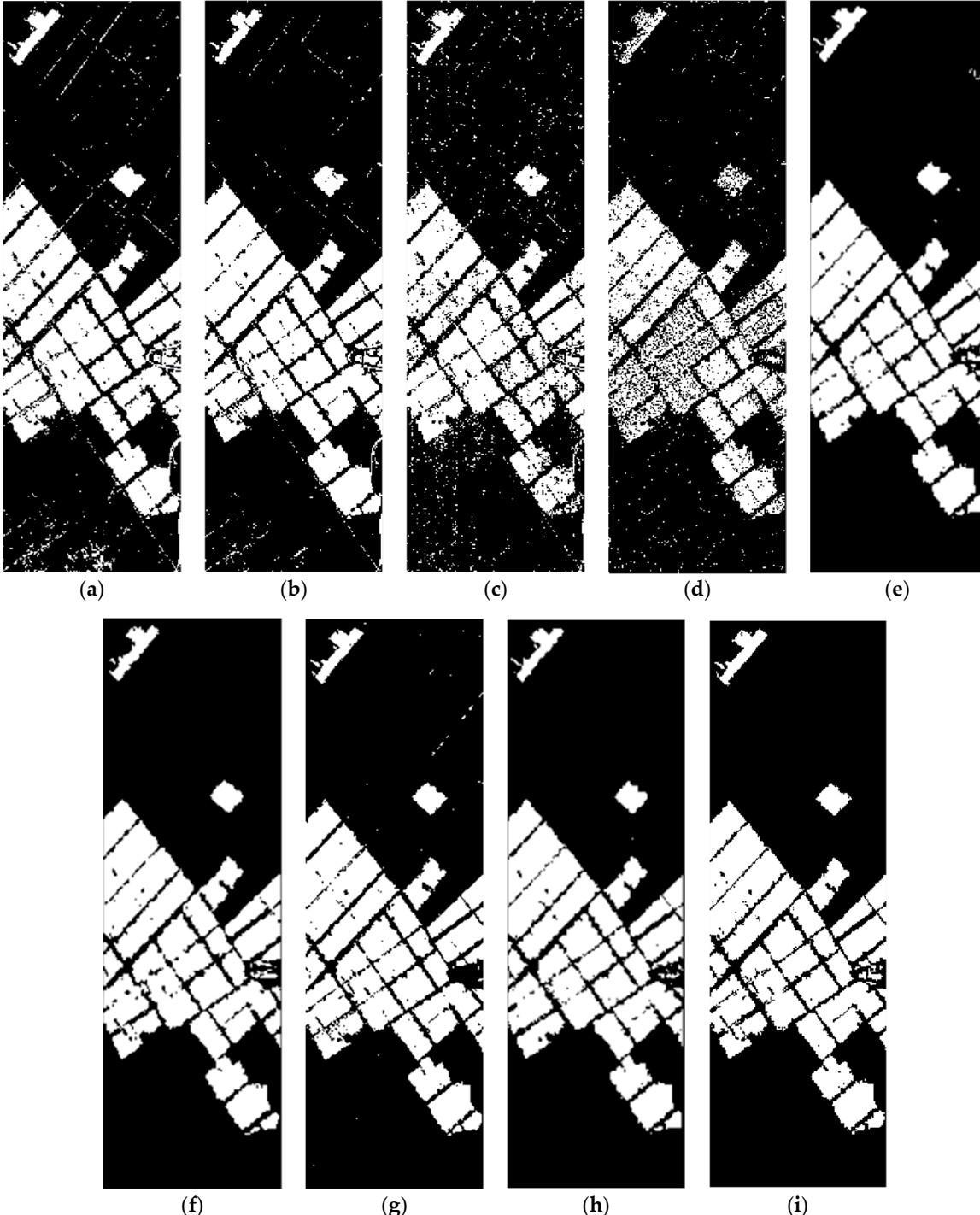

**Figure 4.** Change detection (CD) result on Farmland. (**a**) Change vector analysis (CVA) [14] (**b**) Principal component analysis CVA (PCACVA) [25] (**c**) Support vector machine (SVM) [19] (**d**) Patch-based convolution neural network (PBCNN) [56] (**e**) GETNET [37] (**f**) Hybrid spectral CNN (HybridSN) [41] (**g**) SCNN-S (**h**) SSCNN-S (**i**) Ground truth.

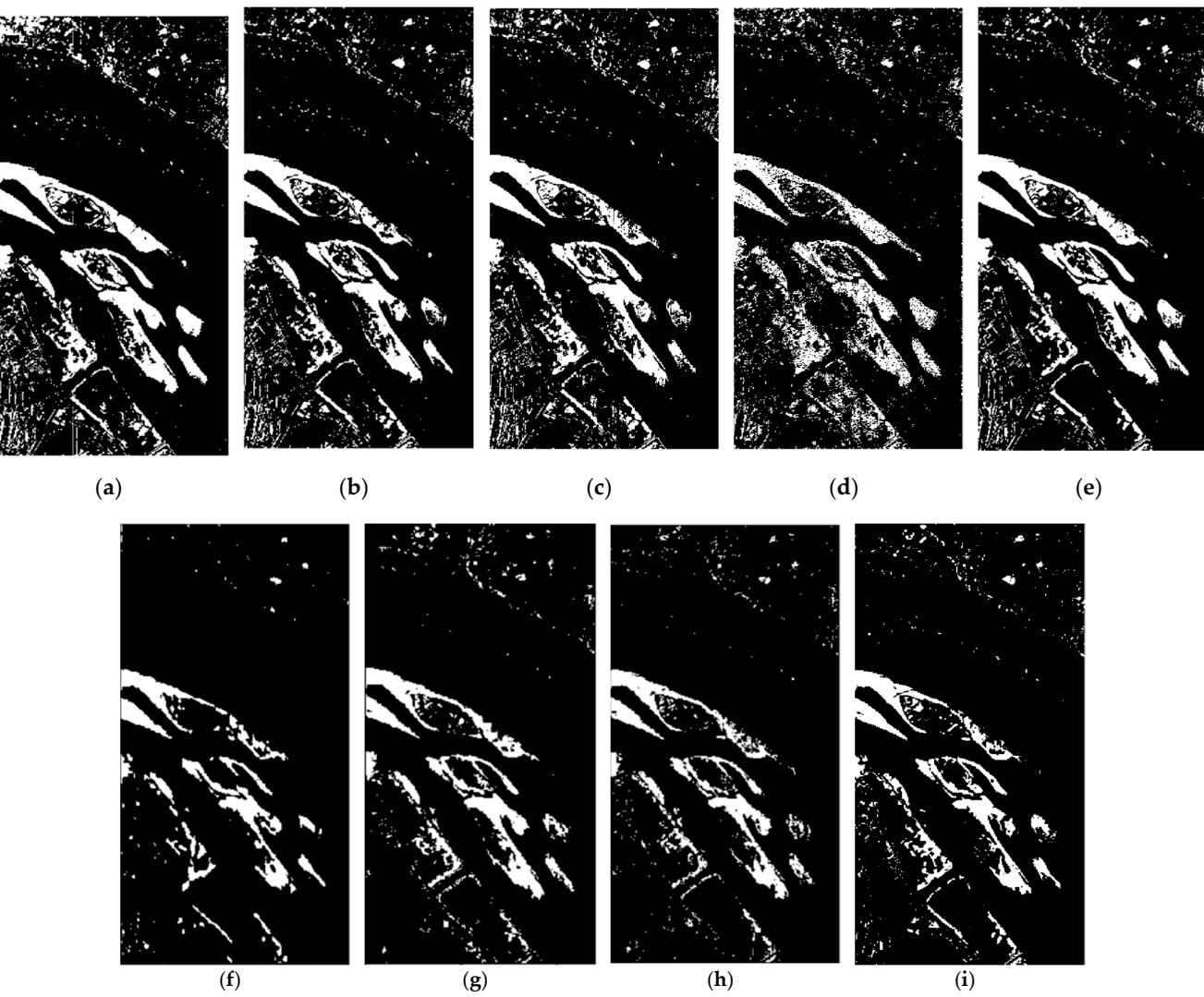

**Figure 5.** CD result on River. (**a**) CVA [14] (**b**) PCACVA [25] (**c**) SVM [19] (**d**) PBCNN [56] (**e**) GETNET [37] (**f**) HybridSN [41] (**g**) SCNN-S (**h**) SSCNN-S (**i**) Ground truth.

**Table 1.** Experimental results of different algorithms on three data sets (the optimal results are highlighted in bold).

| Method | Index | Experiment Data Sets | | |
|---|---|---|---|---|
| | | Farmland | River | USA |
| CVA [14] | OA | 0.9523 | 0.9267 | 0.9200 |
| | Kappa | 0.8855 | 0.6575 | 0.7410 |
| PCACVA [25] | OA | 0.9668 | 0.9516 | 0.9153 |
| | Kappa | 0.9202 | **0.7477** | 0.7225 |
| SVM [19] | OA | 0.9376 | 0.9424 | 0.8810 |
| | Kappa | 0.8483 | 0.7066 | 0.6848 |
| PBCNN [56] | OA | 0.9185 | 0.9139 | 0.8902 |
| | Kappa | 0.7949 | 0.5585 | 0.6699 |

**Table 1.** *Cont.*

| Method | Index | Experiment Data Sets | | |
|---|---|---|---|---|
| | | Farmland | River | USA |
| GETNET [37] | OA | 0.9753 ± 0.0003 | 0.9499 ± 0.0054 | 0.9430 ± 0.0010 |
| | Kappa | 0.9394 ± 0.0008 | 0.7472 ± 0.0215 | 0.8249 ± 0.0030 |
| HybridSN [41] | OA | 0.9749 ± 0.0002 | 0.9614 ± 0.0019 | 0.9553 ± 0.0004 |
| | Kappa | 0.9392 ± 0.0004 | 0.7371 ± 0.0100 | 0.8701 ± 0.0015 |
| SCNN-S | OA | **0.9777 ± 0.0004** | 0.9610 ± 0.0019 | 0.9631 ± 0.0020 |
| | Kappa | **0.9445 ± 0.0012** | 0.7300 ± 0.0069 | 0.8848 ± 0.0059 |
| SSCNN-S | OA | 0.9774 ± 0.0003 | **0.9640 ± 0.0014** | **0.9651 ± 0.0010** |
| | Kappa | 0.9440 ± 0.0004 | 0.7431 ± 0.0034 | **0.8918 ± 0.0022** |

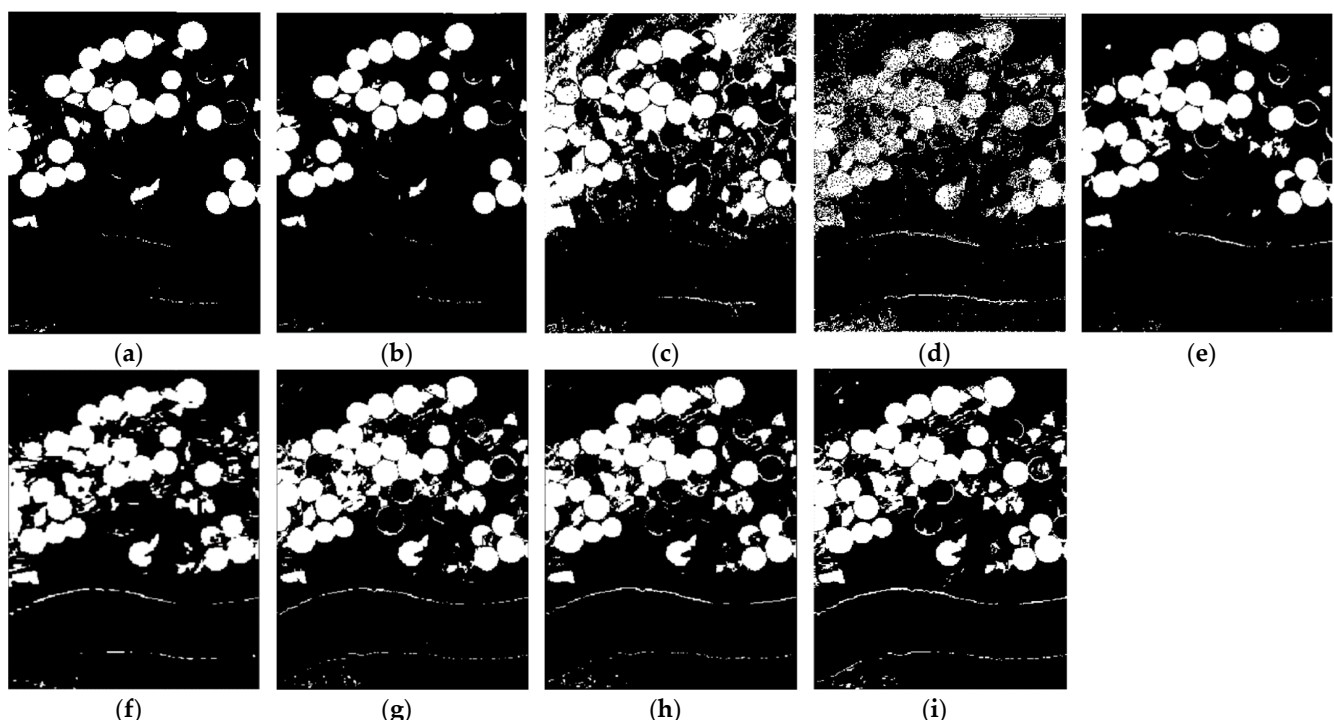

**Figure 6.** CD result on USA. (**a**) CVA [14] (**b**) PCACVA [25] (**c**) SVM [19] (**d**) PBCNN [56] (**e**) GETNET [37] (**f**) HybridSN [41] (**g**) SCNN-S (**h**) SSCNN-S (**i**) Ground truth.

**Table 2.** Reference running time (in seconds) of different algorithms on three data sets.

| Method | Experiment Data Sets | | |
|---|---|---|---|
| | Farmland | River | USA |
| CVA [14] | 29 | 62 | 34 |
| PCACVA [25] | 35 | 67 | 38 |
| SVM [19] | 359 | 86 | 207 |
| PBCNN [56] | 931 | 955 | 1061 |
| GETNET [37] | 515 | 704 | 568 |
| HybridSN [41] | 403 | 311 | 297 |
| SCNN-S | 615 | 288 | 432 |
| SSCNN-S | 925 | 328 | 579 |

## 4. Discussion

### 4.1. Selecting Parameters

4.1.1. Parameters in the Spectral Module

In the spectral module of SSCNN-S, the spatial dimension $b$ of $X_i^{(t)}$ and the dimension $b$ of $ssv_i^{(t)}$ are two important parameters. Moreover, $b$ and $k$ should also be selected differently for different data sets. We used SCNN-S to test different parameter combinations $(b, k)$ on each data sets and observed the Kappa coefficient of CD results to determine which parameter combinations to use. In the parameter experiment, the value range of $b$ was $\{3, 5, 7, 9\}$ and the value range of $k$ was $\{k | k = 15 + 5p, p \in N\}$. Due to SCNN-S also being random, in order to find a Kappa value to represent the CD effect of parameter combination more accurately and quickly, we proposed a four-test scoring method that was analogous to the subjective scoring system of the Chinese College Entrance Examination. For each parameter combination, the four-test scoring method provided a comprehensive Kappa value for that parameter combination after two to four experiments.

---

**Algorithm 2:** Four-test scoring method.

---

**Input:**
1. Error threshold $\vartheta \geq 0$.
**Step 1:** Perform the first experiment and obtain the result $R_1 \in \mathbb{R}$.
**Step 2:** Perform the second experiment and obtain the result $R_2 \in \mathbb{R}$.
**Step 3:** If $|R_1 - R_2| \leq \vartheta$, let the final result be $FR = \frac{R_1 + R_2}{2}$. The algorithm is aborted.
**Step 4:** Otherwise, perform the third experiment and obtain the result $R_3 \in \mathbb{R}$.
**Step 5:** If $|R_1 - R_3| \leq \vartheta$, let the final result be $FR = \frac{R_1 + R_3}{2}$. The algorithm is aborted.
**Step 6:** Otherwise, if $|R_2 - R_3| \leq \vartheta$, let the final result be $FR = \frac{R_2 + R_3}{2}$. The algorithm is aborted.
**Step 7:** Otherwise, perform the final experiment and obtain the result $R_4 \in \mathbb{R}$.
**Step 8:** Let the final result be $FR = \frac{R_1 + R_2 + R_3 + R_4}{4}$.
**Output:**
1. Final result $FR$.

---

In the parameter experiment of this paper, we set $\vartheta = 0.001$.

The results of the Farmland parameter selection experiment using SCNN-S are shown in Figure 7. Although the experimental results of different combinations were somewhat different, they were generally close. Based on the experimental results, we chose a combination of $b = 7, k = 75$ for our experiments on the Farmland data set.

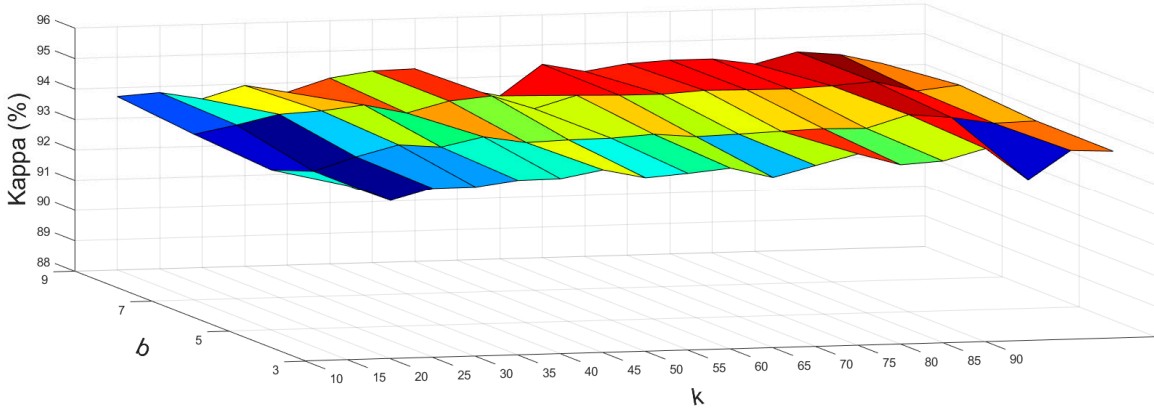

**Figure 7.** Parameter selection experiment for selecting $b$ and $k$ on the Farmland data set.

The results of the River parameter selection experiment using SCNN-S are shown in Figure 8. Based on the experimental results, we chose a combination of $b = 3, k = 85$ for our experiments on the River data set.

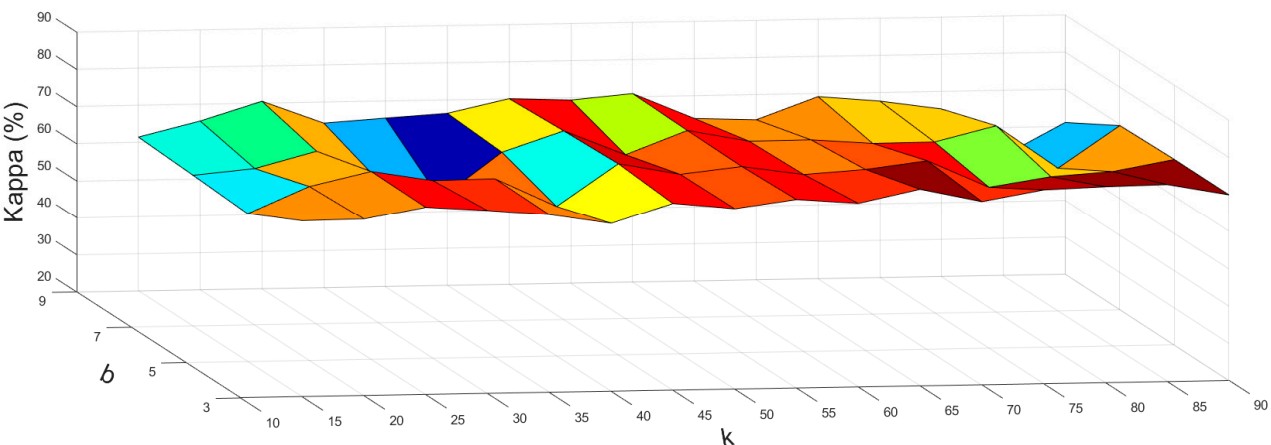

**Figure 8.** Parameter selection experiment for selecting $b$ and $k$ on the River data set.

The results of the USA parameter selection experiment using SCNN-S are shown in Figure 9. Based on the experimental results, we chose a combination of $b = 5, k = 80$ for our experiments on the USA data set.

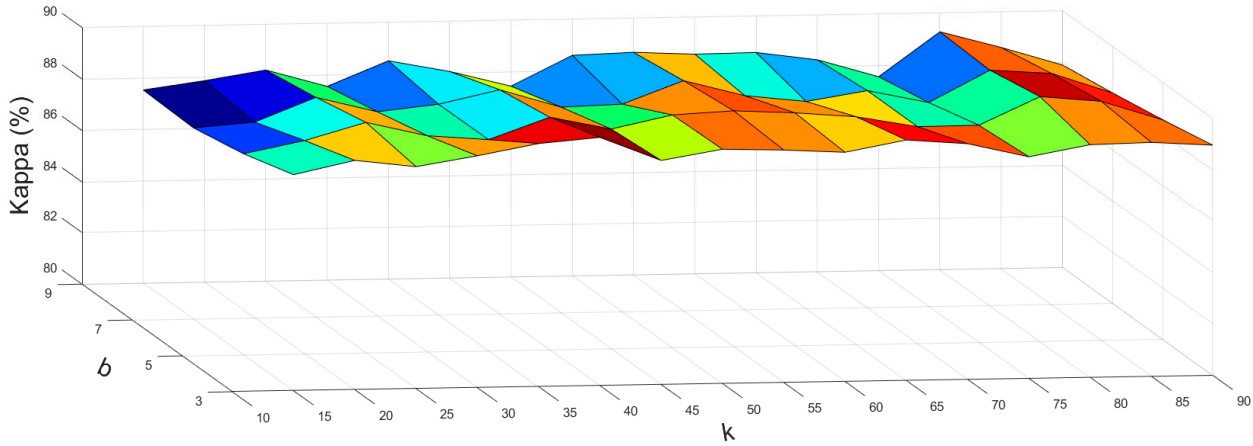

**Figure 9.** Parameter selection experiment for selecting $b$ and $k$ on the USA data set.

### 4.1.2. Parameters in the Spatial Module

In the spatial module, the kernel size of 2D convolution was a parameter that had a greater impact on CD. We conducted the parameter selection experiment of the spatial module on the basis of the results of the parameter selection of the spectral module. The range of the kernel size value $n$ was $\{3, 5, 7, 9, 11\}$. As per the method in Section 4.1.1, we used the four-test scoring method to measure the results of the different parameters.

The experiment results of the parameter selection for selecting the kernel size $n$ of the 2D convolution kernel on the Farmland data set are shown in Figure 10. Based on the experimental results, we chose $n = 9$ for our experiments on the Farmland data set.

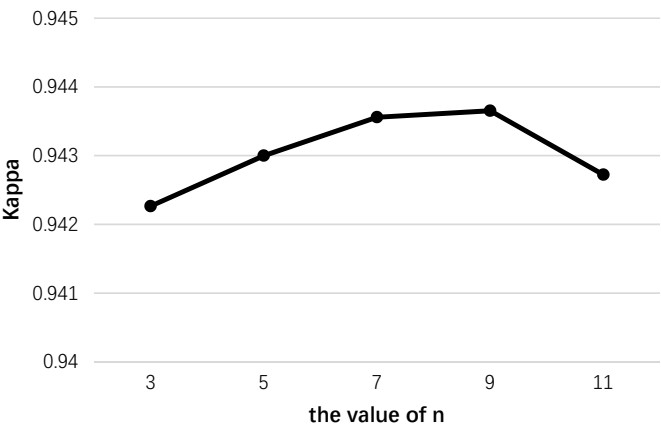

**Figure 10.** Parameter selection experiment for selecting *n* on the Farmland data set.

The experiment results of the parameter selection for selecting the kernel size *n* of the 2D convolution kernel on the River data set are shown in Figure 11. Based on the experimental results, we chose $n = 7$ for our experiments on the River data set.

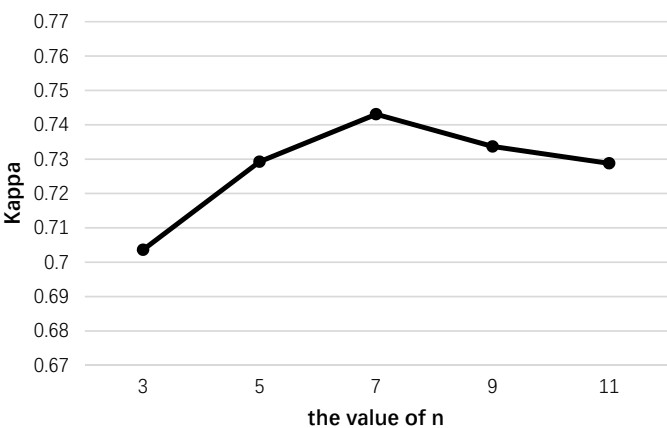

**Figure 11.** Parameter selection experiment for selecting *n* on the River data set.

The experiment results of the parameter selection for selecting the kernel size *n* of the 2D convolution kernel on the USA data set are shown in Figure 12. Based on the experimental results, we chose $n = 9$ for our experiments on the USA data set.

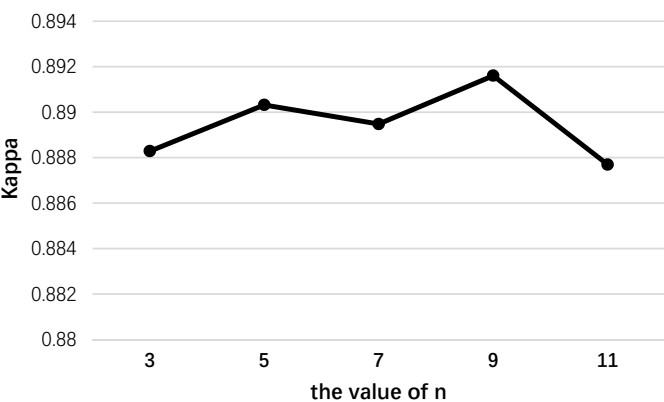

**Figure 12.** Parameter selection experiment for selecting *n* on the USA data set.

*4.2. Discussion on Farmland Experiment*

The Farmland data set was a more standardized data set. The ratio of changed pixels to unchanged pixels was balanced and the changed areas were relatively concentrated and regular in shape. Moreover, there was not a large number of scatter areas. The spectral information was rich and the influence of noise was small. Based on the experiment result, all methods involved in the comparison achieved good results and the main change areas were detected. The CVA, PCACVA, SVM and CNN were all more susceptible to noise, which was evident from the CD results. The HybridSN, through the extracting of spatial features, coped well with situations where different types of changes had similar spectral features. The GETNET incorporated more sub-pixel information than a CNN. SCNN-S and SSCNN-S extracted spectral features through a deep network that greatly reduced the impact of noise and achieved excellent CD results.

*4.3. Discussion on River Experiment*

The River data set was very challenging. Based on the ground truth map, it was evident that the River data set had a wide variety of variations, irregular areas of variation and a large number of scatters. Additionally, the ratio of changing pixels to total pixels in the River data set was less than 10%. This extremely unbalanced data set made classifying the changed into the unchanged greatly affect the Kappa coefficients. Based on the experiment results, SSCNN-S achieved the best accuracy and PCACVA achieved the best Kappa coefficient, which was similar to the results of [26]. This result demonstrated that SSCNN-S was more stringent in judging changed pixels while spatial information supplementation effectively improved the accuracy and Kappa value compared with SCNN-S. Overall, the Kappa coefficients for all methods were low, which demonstrated the high complexity of this data set.

*4.4. Discussion on USA Experiment*

The USA data set was a more comprehensive data set that contained a lot of circular change areas and a few curve and scatter areas. Based on the experimental results, the area of curve change representing the river boundary was a watershed that affected the detection results of the different methods. Of all of the methods involved in the comparison, only SCNN-S and SSCNN-S depicted changes in the river boundary very well, indicating that the spectral module could extract the spectral features of the tensor more effectively. Compared with SCNN-S, SSCNN-S significantly improved the OA and Kappa coefficients, indicating that the auxiliary features of the spatial module made up for the noise interference of the spectral features.

**5. Conclusions**

In this paper, we proposed a spectral-spatial convolution neural network with Siamese architecture (SSCNN-S) for CD. SSCNN-S extracted the spectral-spatial features of hyperspectral tensors through spectral and spatial modules and converted the hyperspectral tensors into spectral-spatial vectors. SSCNN-S introduced contrastive loss into CD and transformed the CD problem into a similarity measurement problem of spectral-spatial vectors. The distance function was used to calculate the distance between two spectral-spatial vectors to describe the similarity of two tensors and this was used as the basis of CD. Considering the high spectral resolution and low spatial resolution of the HSIs, SSCNN-S used a spectral-spatial combination method to extract features of the hyperspectral tensor, which mainly relied on the spectral features and supplemented the spatial features. Based on the full extraction of spectral features, the auxiliary spatial features weakened the influence of noise on the spectral features. The introduction of the Siamese network architecture into CD bridged a gap between hyperspectral classification and CD. A few advanced hyperspectral classification techniques could be applied to CD based on this architecture. From the experience, SCNN-S and SSCNN-S achieved good results on three real data sets.

In our future work, we will continue to explore the following two aspects. The first aspect is the HSI. In addition to the real HSI, the use of synthetic data may bring further discoveries. The second aspect is the further research on the spatial module. We tested the effect of different sizes of 2D convolution kernels on the results in our experiments. In fact, the spatial module can have more architectures to extract the spatial features of HSIs as fully as possible.

**Author Contributions:** Conceptualization, T.Z., B.S., Y.X. and Z.W.; methodology, M.W. and X.W.; software, B.S. and G.Y.; validation, T.Z., B.S. and Y.X.; formal analysis, M.W.; investigation, M.W.; resources, G.Y.; data curation, B.S.; writing—original draft preparation, T.Z. and B.S.; writing—review and editing, B.S.; visualization, B.S.; supervision, T.Z.; project administration, T.Z. and Z.W.; funding acquisition, T.Z. and Z.W. All authors have read and agreed to the published version of the manuscript.

**Funding:** This research was funded by the National Natural Science Foundation of China under Grant 61976117, 62071233, 61876213, 61772274 and 61772277 in part by the Natural Science Foundation of Jiangsu Province under Grant BK20191409, BK20201397, BK20180018 and BK20171494, in part by the Key Projects of University Natural Science Fund of Jiangsu Province under Grant 19KJA360001 and 18KJA520005, in part of the National Key R&D Program under Grant 2017YFC0804002, in part by the Fundamental Research Funds for the Central Universities under Grant 30917015104, 30919011103 and 30919011402, in part by the Collaborative Innovation Center of Audit Information Engineering and Technology under Grant 18CICA09, in part by the Young Teacher Research and Cultivation Project of Nanjing Audit University under Grant 18QNPY015 and in part by the Postgraduate Research and Practice Innovation Program of Jiangsu Province under Grant KYCX20_1680.

**Acknowledgments:** We are grateful to Wang Qi and Hasanlou Mahdi who provided the data for this research.

**Conflicts of Interest:** The authors declare no conflict of interest. The funders had no role in the design of the study; in the collection, analyses or interpretation of data; in the writing of the manuscript or in the decision to publish the results.

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
