# Peer review of "SSCNN-S: A Spectral-Spatial Convolution Neural Network with Siamese Architecture for Change Detection"

_remotesensing, doi:10.3390/rs13050895_

Round 1

Reviewer 1 Report

The submitted work proposed a spectral-spatial network with Siamese architecture to solve the problem of hyperspectral binary change region detection. It’s mainly based on the extraction of tensor pairs taken from HSIs recorded at different time-points. The proposed network combines spectral and spatial information to reduce the impact of noise. A new scoring method is also suggested (instead of the common mean value) to improve the performance. Generally speaking, the goal of the paper is clear and interesting to the readers, as change detection and monitoring are well-discussed but still challenging problems in remote sensing. The novelty of the proposed work seems acceptable as several modifications to the available networks/approaches have been done. On the other hand, the structure of the manuscript is fine but the language needs to be thoroughly improved. Finally, after carefully reviewing the manuscript, I believe the manuscript needs to be revised before being considered for probable publication. I would suggest addressing the comments (which are briefly listed as below) and resubmitting the revised version of the manuscript.

  • The proposed method has been compared to two old approaches (i.e based on CVA [Malila et al., 1980], and support vector machine [Nemmour et al., 2006]) as well as two newer methods (which are patch-based CNN (PBCNN) and GETNET [Wang et al., 2018]). I suggest adding at least one other state-of-the-art method to the comparison section.
  • Moreover, assuming that the performance of the proposed deep learning method is better than applying traditional methods such as SVM and CVA (in terms of accuracy, sensitivity, etc.), it would not be necessarily enough to encourage the user to utilize the DL-based method rather than the traditional ones. One should notice that the complexity of a method like a traditional SVM is not comparable with that of a relatively complex deep CNN. Then I suggest adding a section on computational complexity and/or processing time and compare the methods in terms of their complexity.
  • Following the comment above, it would be helpful if you add some information about the system which is used for running the network (I mean some info regarding the used CPU/GPU/RAM, etc.)
  • Line 15: in the sentence: ‘…in two HSIs separately..’ , which ‘two’ HSIs? I suggest clearly stating that you are talking about the HSIs recorded at two different time-points.
  • Please add references next to the methods compared in Table 1 and Table 2. Drawing horizontal lines between different methods could be also helpful to separate them.
  • I would have the same suggestions for figures in which different methods are being compared.
  • Although I am not insisting on this, the term ‘Change Region Segmentation‘ does not sound that common and familiar, therefore it can be substituted with some more common and widely-used terms such as change detection and so on, as also called like this in almost all the other related literature.
  • In general, although the paper is still easy-to-follow, there are several typos and grammatical mistakes here and there. Please give a closer look at it.
  • Line 307: The sentence seems impaired.
  • Please define parameters k and b whether in the caption of Figure 7 or in the figure itself.
  • Page 12: Same suggestion for Figure 8.
  • Please use the same font size and style throughout the text. For example, numbers in Line 260 and Line 272 stating the spatial size are different in size in comparison to the other numbers.

Author Response

Thank you very much for your comments concerning our manuscript entitled “SSCNN-S:a Spectral-Spatial Convolution Neural Network with Siamese architecture for Change Detection” (ID: Remote Sensing-1088594). Those comments were all extremely valuable and very helpful for revising and improving our paper, as well as for providing important insights and significance to our research. We have studied the comments carefully and have made corrections to our manuscript which we hope will meet your approval.

Please find below file which is our description of the changes made to the manuscript according to comments received (those changes have been highlighted in red color in the revised version of the manuscript in order to facilitate their identification with regards to the previous version). We are indebted to the Editors and Reviewers for their constructive comments and suggestions, which greatly helps us to improve the technical quality and presentation of our manuscript.

Reviewer 2 Report

In this paper a Siamese DNN method is presented for CRS. First all pixels create a bxbxc subdomain and pairs from the two HSIs are fed in the network. At first a Neural sub Network is applied on the depth (spectrum), what they call 1D convolution  then after the dimensionality of the spectrum is reduced it is fed into a spatial submodule and a final vector is produced. The pairs are compared with an Eucledian norm and they drive the training this way on the siamese network.   

One of my concerns is that they force the field of view to be 3x3, I do not like this. I totally agree that it is a good strategy for memory concerns to have a bxbxc tensors rather than having the bulk of the hyperspectral image. Wouldn't it be ok to play a bit with the field of view and let some pooling also happen? I think there are several architectures that can be tested.  

What I am though really opposed in this paper is that they use just two HSIs thus they overfit (since I think the parameters are few) their network for their pair of HSIs. They are really starting nice with the bxbxc which is the "cropping" of the image for memory issues. But they really mess up for me having just two HSIs. They should have many, unless they can prove that two HSIs are enough for generalizing the problem to all HSIs which I doubpt...so please use many and resubmit again. 

Author Response

(The authors gave the same response as above.)

Round 2

Reviewer 1 Report

After a careful comparison of the current version with the previous version of the manuscript, I believe the quality of the paper is significantly improved after addressing the concerns of the reviewers. Therefore, I believe the re-submitted work deserves to be published. However, I still suggest proofreading the text in order to ensure a flawless publication.

Author Response

Thank you very much for your positive comments. We have carried out proofreading to ensure that what we want to express is accurate.

       We gratefully thank the reviewer again for his/her outstanding comments and suggestions, which greatly helped us to improve the technical quality and presentation of our manuscript.

Reviewer 2 Report

I think synthetic data can maybe produce better generalization. I will let the paper pass but my reluctancy remains in this review session. Just because many do something in not a good way does not make it good...

About the kernel size. I would have also tried a hierarchical approach starting for example with n=3, n=5 and n=7. If you do try this and achieve better results put this in the paper, I do not need to accept it. I expect it to be better though.

So let me pass the paper with the hope the researchers will further improve their approach or researchers will find some ideas inspiring and improve it. 
Generally though I must say that seeing their approach would tempt me to experiment also. So yes I agree to pass it.

Author Response

Thank you very much for your comments concerning our manuscript entitled “SSCNN-S:a Spectral-Spatial Convolution Neural Network with Siamese architecture for Change Detection” (ID: Remote Sensing-1088594). Those comments were all extremely valuable and very helpful for revising and improving our paper, as well as for providing important insights and significance to our research. We have studied the comments carefully and have made corrections to our manuscript which we hope will meet your approval.
